# HeteEdgeWalk: A Heterogeneous Edge Memory Random Walk for Heterogeneous Information Network Embedding

**DOI:** 10.3390/e25070998

**Published:** 2023-06-29

**Authors:** Zhenpeng Liu, Shengcong Zhang, Jialiang Zhang, Mingxiao Jiang, Yi Liu

**Affiliations:** 1Information Technology Center, Hebei University, Baoding 071002, China; lzp@hbu.edu.cn; 2School of Cyber Security and Computer, Hebei University, Baoding 071002, China; zhangshengcong5@gmail.com (S.Z.); liangzi990613@gmail.com (J.Z.); mingxiaojiang11@gmail.com (M.J.)

**Keywords:** network embeddings, random walk, heterogeneous information network, representation learning, edge sampling

## Abstract

Most Heterogeneous Information Network (HIN) embedding methods use meta-paths to guide random walks to sample from HIN and perform representation learning in order to overcome the bias of traditional random walks that are more biased towards high-order nodes. Their performance depends on the suitability of the generated meta-paths for the current HIN. The definition of meta-paths requires domain expertise, which makes the results overly dependent on the meta-paths. Moreover, it is difficult to represent the structure of complex HIN with a single meta-path. In a meta-path guided random walk, some of the heterogeneous structures (e.g., node type(s)) are not among the node types specified by the meta-path, making this heterogeneous information ignored. In this paper, HeteEdgeWalk, a solution method that does not involve meta-paths, is proposed. We design a dynamically adjusted bidirectional edge-sampling walk strategy. Specifically, edge sampling and the storage of recently selected edge types are used to better sample the network structure in a more balanced and comprehensive way. Finally, node classification and clustering experiments are performed on four real HINs with in-depth analysis. The results show a maximum performance improvement of 2% in node classification and at least 0.6% in clustering compared to baselines. This demonstrates the superiority of the method to effectively capture semantic information from HINs.

## 1. Introduction

In this current era of rapid Internet development, the volume of data is rising, and the correlation of data is becoming more complex. Most of them are presented as networks with social relationships and genome, academic, and protein interactions, to name a few. Network embedding has become a major research direction. The main concept of mapping the nodes in the network is to a consecutive lower dimensional vector representation. It is utilized to preserve key structural and semantic information in the network. Not only that, network embeddings can be applied to all kinds of supervised and unsupervised downstream tasks, such as node classification [1,2], link prediction [3,4], and node clustering [5,6], among others. 

Random walks are extensively used for exploring information about nodes in the immediate vicinity of a network [7,8]. These methods can be used to represent nodes in homogeneous information networks; they are not appropriate for HINs. HINs are more complex, with multiple types of nodes containing both homogeneous and heterogeneous edges. A simple academic HIN is depicted in Figure 1a, consisting of three different types of nodes, three heterogeneous edges, and one homogeneous edge. The nodes are author (A), paper (P), venue (V), and topic (T), while the heterogeneous edges are A–P, P–T, and P–V, respectively. Inspired by homogeneous information network methods, Dong et al. and Shi et al. [9,10] proposed to sample network nodes via meta-path guided random walks to solve the complex relationships of HINs. As shown in Figure 1b, just as the meta-path A–P–A in [9] represents two authors co-authoring a paper, A–P–V–P–A represents two authors publishing papers in the same venue. However, the selection of the best meta-paths continues to present a difficult time and quality challenge. A single meta-path is not adequate to represent the rich structural and semantic information in HINs. Additionally, the choice of meta-paths frequently calls for specialized domain knowledge. Fu et al. [11] propose a strategy using automatic extraction of meta-paths below a threshold value. However, the quantity of these self-selected meta-paths increases exponentially with their length. In any case, the quality of the predefined meta-path or the length chosen have a significant impact on the outcome. Network schemas are used by [12] to guide their spatial random walks. As shown in Figure 1c, network schema is defined as a minimal graph consisting of all node and edge types in a HIN together. However, the results of the sampling will be biased in favor of types where there appear to be more nodes. Samy et al. [13] also use network schema to guide random walks, balancing the choice of edge types in the network. However, the graph data itself is unevenly distributed. Nodes of the same type accumulate together, leading to starvation. This makes some of the clustered nodes inaccessible, losing node information.

Network embedding methods depend on the quality of sampling. Both homogeneous and heterogeneous network embedding methods improve the quality of sampling as much as possible to produce a better representation of the network structure and semantic information. The main problem is that the nodes with high centeredness in the network may appear more frequently in the walk sequence. This makes the sampling unbalanced, which leads to a negative impact on the embedding results. The main concern of this paper is how to choose the rich information in HINs in a balanced manner while minimizing the influence of the central node on the sampling results. We propose HeteEdgeWalk, a random walk that samples flexibly through network schema, to achieve this. It mitigates bias when exploring HINs by regulating random walk strategy. To be precise, our aim is a more balanced sampling of edge types. We use a dynamically adjusted bidirectional edge sampling walk strategy. Specifically, we make a probabilistic selection of the next type of edge in the walk by means of an exponential decay function. It reduces the frequency of recurrence of the same edge type in the random walk. Then, we use a dynamic resizing sequence to store the last few edge types to mitigate the probability of starvation during the walk. 

The main contributions of our work are as follows: We propose HeteEdgeWalk, a HIN-embedding technique based on network schema. It is an approach that does not involve meta-paths and optimizes the representation more fairly and comprehensively by capturing composite interactions;We analyze the disadvantages of sampling based on node types and suggest sampling based on edge types as a more versatile and finer-grained method for exploring HINs;Four large real-world datasets are utilized to evaluate the experimental. The results show that HeteEdgeWalk achieves optimal or sub-optimal performance compared to the baselines. In the study of parameter sensitivity, our approach exhibits robustness in various scenarios.

## 2. Related Work

According to the type of graph, homogeneous and heterogeneous information networks are the two major categories of available graph-embedding methods. On the one hand, all nodes of a homogeneous information network belong to a single domain, such as a social network of users. Many graph-embedding methods attempt to use the adjacency matrix to obtain the embedding representation of homogeneous information network. However, the sparsity of the adjacency matrix and the amount of matrix computation posed significant challenges. Perozzi et al. [7] proposed a strategy to learn node embeddings by combining a random walk with the SkipGram model [14]. SkipGram maximizes the co-occurrence probability of nodes in context window in random walks. Tang et al. [8] achieved better embedding by preserving first and second order proximity in homogeneous information networks. On the other hand, HINs exhibit a more complicated architecture. Its nodes belong to various domains with heterogeneous and homogeneous edges. The homogeneous information network embeddings directly extending to HINs will result in low-quality node embeddings. The performance of downstream graphics analysis tasks may be further degraded as a result. Dong et al. [9] propose a combination of meta-path guided random walks and heterogeneous SkipGram strategy to capture the key structural properties of HINs. They manually select A–P–A and A–P–V–P–A as meta-paths of the academic HIN in accordance with a priori knowledge. However, the selected meta-path is only appropriate for this network. This causes the meta-path to be highly dependent on the expertise of domain experts.

Some methods either select meta-paths according to predefined criteria or use other approaches instead of meta-paths to achieve improved robustness. First, the specific criterion is generally to set a threshold value for the length of meta-paths, and only those below the threshold are retained. Fu et al. [11] combine all meta-paths below a predetermined threshold in order to direct random walks and thus learn different relationships between nodes. Second, other methods can also be used instead of meta-paths. Zhang et al. [15] use meta-graph to direct random walks. However, the generation of the meta-graph is a considerable overhead. Hussein et al. [16] propose the Jump and Stay strategies to regulate the sampling of node sequences. The Stay strategy keeps the current domain sampled and saves the most recent node types so that domains with the same node type are not visited repeatedly in the near future, as opposed to the Jump strategy, which selects to leap to the next domain with a different type of node. This successfully strikes a slight balance between sampling at heterogeneous and homogeneous edges. However, it is challenging to access short sequences, such as (A–P–A), which leads to the loss of node information. He et al. [12] used network schema to guide their spatial random walks. The results of the sampling, however, will be biased in favor of types where there appear to be more nodes. Samy et al. [13] also used network schema to guide random walks, balancing the choice of edge types in the network. However, the graph data itself are unevenly distributed, with nodes of the same type accumulating together. This leads to starvation, thus making some of the clustered nodes inaccessible, losing node information. In this paper, we propose a flexible sampling strategy that does not utilize any meta-path. It can not only provide a balance of heterogeneous edges but also mitigates the probability of starvation during the walk. Our edge-sampling method can capture the HIN information better, as compared to node sampling. It mitigates the effect of high-frequency nodes in random walks and provides a more balanced sampling of heterogeneous edges.

Finally, there are other HIN-embedding techniques, such as the use of deep learning frameworks [17], hyperedges [18], hypergraphs [19], and HeGAN [20], which learn node embeddings using adversarial learning. In addition to static charts, there are also technologies applied to dynamic graphs [21]. In terms of applications, HIN embedding can be applied to many aspects, such as recommender systems [10], business networks [22], and bioinformatics networks [23].

## 3. Materials and Methods

In this section, we provide a detailed description of our HeteEdgeWalk method. It utilizes a dynamically adjusted bidirectional edge sampling walk strategy on a HIN without involving meta-paths. Specifically, HeteEdgeWalk first conducts random walks on the imported HIN. In selecting the next node, our method first performs probabilistic balanced edge sampling. Then, we randomly select nodes by sampled edge types. Finally, these random walks are input into the SkipGram to output node embeddings. Hereafter, we first introduce some key concepts. Subsequently, we describe in detail dynamically adjusted bidirectional edge sampling walks and node embedding learning with SkipGram. Finally, the materials and experimental setup are described.

### 3.1. Preliminaries

Heterogeneous Information Network (HIN). Given as a graph G=(V, E, A, R), where V and E represent set of nodes and edges, respectively. Given a function ϕ: V→A for node-type mapping and a function ψ: E→R for edge-type mapping, A and R represent node-type and edge-type set, respectively. When $\left|A\right|+\left|R\right|>2$ is a HIN, the opposite is a homogeneous information network. 

Heterogeneous Information network embedding. Given a HIN G=(V, E, A, R), for each node v∈V, the aim is to learn the *d*-dimensional consecutive embedding function $f:\;V\to R^{d}$ to maintain each node v semantic and structural information in the HIN, where *d*
$\ll \left|V\right|$.

Network schema. Given a HIN G=(V, E, A, R), a network schema is a minimal graph $T_{G}=(A,R)$ containing all node and edge types.

### 3.2. Dynamically Adjusted Bidirectional Edge Sampling Walk

We design dynamically adjustable edge-sampling strategy without meta-path guidance inspired from [16]. Jiang et al. proposed bidirectional random walks, which allow retrieval of more tail nodes than a unidirectional one, in theory [24]. We have followed this idea to design a bidirectional edge sampling walk. After this, the bidirectional edge-sampling walk is simply called Bi-EdgeSamplingWalk. It is an edge-sampling strategy that adjusts for walks by rationalizing the choice of edges. To guide the random walk behavior in the HIN, the following probability is first defined to select the next edge type $r_{i+1}$: (1)$$\begin{array}{c} p\left(r_{i+1}|v_{i}\right)=\left\{\begin{array}{l} Weight\left(r_{i+1}\right),r_{i+1}\in R_{N}\left(v_{i}\right)\\ 0,\;\mathrm{otherwise} \end{array}\right. \end{array}$$
where $R_{N}(v_{i})$ denotes the set of all edge types connected to the current node $v_{i}$, and Weight(r) denotes the probability of all edge types being selected.

We adjust Weight(r) by the number of visits n to this edge type. The probability decreases progressively as the number *n* increases, as follows: (2)$$Weight\left(r\right)=\alpha^{n},\alpha \in \left(0,\;1\right)$$
where α is the decay parameter and is set to 0.4. 

Node types that link multiple-type edges in the graph often act as pivots. To reduce the influence of this high-frequency type of nodes, we sample the different edge types as evenly as possible. A queue Q is designed for storing the most recently visited edge types, with the following rules: (3)$$\begin{array}{c} \left\{\begin{array}{l} in\left(r\right),\;\mathrm{if}r\notin Q,\left|Q\right|<\left|R_{N}\left(v_{i}\right)\right|/2\\ pop()\mathrm{and}in\left(r\right),\mathrm{if}r\notin Q,\left|Q\right|\geq \left|R_{N}\left(v_{i}\right)\right|/2\\ pop\left(r\right)\mathrm{and}in\left(r\right),\;\mathrm{if}r\in Q \end{array}\right. \end{array}$$
where in(r) represents the in-queue operation, which adds edge type r to the end of queue Q. The pop() represents the out-of-queue operation. The pop(r) means to remove the edge type r from the queue Q. When the queue $\left|Q|=|R_{N}(v_{i})\right|$, all edge types connected by the current node $v_{i}$ have been visited in the recent past. After that, $r\in R_{N}(v_{i})$ is chosen randomly as the next edge type to achieve more balanced edge type access. 

Afterwards, the next node of the present node $v_{i}$ is selected depending on the edge type, as follows: (4)$$\begin{array}{c} p\left(v_{i+1}|v_{i},r_{i+1}\right)=\left\{\begin{array}{l} \frac{1}{\left|N_{r}\left(v_{i}\right)\right|},r_{i+1}\in R_{N}\left(v_{i}\right)\\ 0,\;r_{i+1}\notin R_{N}\left(v_{i}\right) \end{array}\right. \end{array}$$

The above process is the basic strategy of our algorithm. To better understand this process, Figure 2 illustrates an instance of a walker in a simple academic HIN with Bi-EdgeSamplingWalk. 

It shows the storage queue and weight changes in a walker with A1 as the starting node. Set α and weights to 0.1 and 1, respectively. Taking the left half as an example, nodes of type A are linked to nodes of type P only, and A1 can only select edges of type A–P. Weight (A–P, I = 1) = 0.1, where I is the number of selections. Meanwhile the queue is initially empty, and the $in\left(r\right)$ operation is performed directly. After walking to P1, P links four edge types, all with a weight of 1. P–T is randomly chosen as the next edge type and the weights are updated. Because $\left|Q\right|=1<\left|R_{N}\left(v_{i}\right)\right|/2=2$, the $in\left(r\right)$ operation is performed and the queue Q stores A–P and P–T. After walking to T1, $\left|Q\right|\geq \left|R_{N}\left(v_{i}\right)\right|/2$, the $pop()\;\mathrm{and}\;in\left(r\right)$ operation is performed, the queue Q is updated to P–T and T–P. Meanwhile, the weight of T–P is updated to 0.1. After walking to P3, since P–T has a weight of 0.1 and the other three edge types have a weight of 1, P–P is chosen randomly as the next edge type and the weights are updated. $\left|Q\right|=\left|R_{N}\left(v_{i}\right)\right|/2$, the queue Q is updated to T–P and P–P.

The pseudo-code for the Bi-EdgeSamplingWalk algorithm is displayed in Algorithm 1.


**Algorithm 1:** Bi-EdgeSamplingWalk (G, v,l, α)  **Input:** the graph G=(V, E, A, R), the start node per walk v, the walk length l, the decay parameter *α*.**Output:** the random walk sequence W.1: Initialize W;2: W=[v];3: vf=vb=v;4: αf=αb=α;5: **for**
i=1→[l/2] **do**6:       drawvfaccording to Equations (1)~(4);7:       drawvbaccording to Equations (1)~(4);8:       update αf and αb;9:       W=[vb]+W+[vf];10: **end for**11: return W


### 3.3. Node Embedding Learning with SkipGram

After obtaining the random walk sequence W, HeteEdgeWalk uses the SkipGram [14]. In particular, it aims to maximize the co-occurrence probability p of nodes that appear in the identical context window k, as follows:(5)$$\underset{\theta }{\mathrm{argmax}}\sum_{w\in W}\sum_{v\in w}\sum_{c\in N\left(v\right)}\log p\left(c|v;\theta \right)$$
where w is a walk and $N\left(v\right)$ is the neighborhood nodes in w. The node v does not exceed the size of context window k. The co-occurrence probability is generally defined as a softmax function, as follows: (6)$$p\left(c|v;\theta \right)=\sigma \left(\vec{v}\cdot \vec{c}\right)=\frac{exp\left(\vec{v}\cdot \vec{c}\right)}{\sum_{u\in V}exp\left(\vec{v}\cdot \vec{u}\right)}$$

The number of nodes is usually large. The negative sampling technique [25] is generally used to achieve an approximate probability. It maximizes the probability that the target node does not occur at the same time as a randomly sampled negative node. The final maximization objective is formulated as follows:(7)$$\log\left(\vec{v}\cdot \vec{c}\right)+\sum_{m=1}^{M}E_{u\sim p\left(u\right)}[\log\sigma \left(-\vec{v}\cdot \vec{u}\right)]$$
where M is the number of negative samples and $p\left(u\right)$ is the distribution of negative samples. The above equation is the full process for SkipGram to learn the node representation in HIN.

Finally, the pseudo-code for the entire HeteEdgeWalk algorithm is given in Algorithm 2. HeteEdgeWalk as a whole is divided into two parts, Bi-EdgeSamplingWalk and SkipGram. In the first part, we perform edge sampling and generate random walks on the HIN. When the next node is to be selected, our method first performs probabilistic balanced edge sampling by parameter α. Then, the most recently visited edge types are stored according to certain conditions. Finally, we use the sampled edge types to randomly select nodes. In the second part, we input the random walks into the SkipGram to learn the node embedding results.


**Algorithm 2:** HeteEdgeWalk (G,λ, l,d,k, α)   **Input:** the graph G=(V, E, A, R), the numbers of walks per node λ, the walk length l, the node-embedding dimension d, the context window size k, the decay parameter *α*.**Output:** the node representations X∈R|V|×d1: Initialize X;2: for i=1→λ do;3:   **for**
v∈V **do**;4:    *W* = Bi-EdgeSamplingWalk (G, v,l, α)5:    X=SkipGram(X, W, d, k);6:   **end for**7: **end for**8: return X


### 3.4. Experimental Setup

#### 3.4.1. Dataset Description

We use four real heterogeneous networks for evaluation: DBLP [13], ACM [26], Foursquare [27], and Movies [13]. Table 1 contains statistics on the four HINs. 

DBLP [13] is an academic network based on four types of nodes. It includes 5915 authors (A), 5237 papers (P), 4479 topics (T), and 18 venues (V). In addition, each author corresponds to the below four domain areas: “data mining”, “information retrieval”, “database”, and “machine learning”;ACM [26] is the other academic network containing three types of nodes. The nodes of the network include 7167 authors (A), 4039 papers (P), and 60 subjects (S). Each paper falls under one of the three research categories of “databases”, “wireless communications”, or “data mining”;Foursquare [27] is a graph with four different types of nodes based on the user’s check-in history in New York City. It contains 1250 points of interest (P), 2449 users, 25,904 check-ins (C), and 168 timestamps (T). Additionally, each point of interest (P) is given a label in accordance with its class, such as “bar”;Movies [13] is an augmented movie graph with four different types of nodes. There are 10,350 actors (A), 6517 movies (M), 2582 directors (D), and 1335 composers (C) in it. The original data contains only three heterogeneous edge types (M–A, M–D, and M–C), while the augmented data adds two homogeneous edges (M–M and A–A). In addition, each movie corresponds to the following movie themes: “action”, “horror”, “adventure”, “science fiction”, and “crime”.

#### 3.4.2. Baselines

We conduct a comparison with five current state-of-the-art random walks in homogeneous and HINs, including DeepWalk [7], Metapath2Vec [9], JUST [16], Bidirectional Random Walks [24], and SchemaWalk [13]. 

DeepWalk [7] is a classical homogeneous graph-embedding algorithm. It first performs random walks for each node and transforms the obtained random walks into node embeddings using the SkipGram model;Metapath2Vec [9] is a HIN-embedding algorithm based on meta-path-guided random walks. It first guides random walks through meta-paths defined in advance and then learns node embedding through the SkipGram model. For DBLP, the meta-paths as in [13] are P–A–P and A–P–V–P–A. The selection of P–A–P and P–S–P as meta-paths for ACM is similar to the selection in [26]. Similar to [13], the meta-paths for Foursquare are U–C–P–C–U and P–C–T–C–P, and the meta-paths for Movies are A–M–D–M–A and A–M–C–M–A, respectively;JUST [16] is a HIN-embedding algorithm that proposes whether meta-paths are necessary or not. The relational information of each node is treated as a domain, and whether to jump to the next node domain is controlled by the Jump and Stay operations. Jump is equivalent to selecting heterogeneous nodes, and similarly, Stay selects homogeneous nodes. It prioritizes heterogeneous Jump and reduces the heterogeneous Jump probability to achieve balanced sampling of heterogeneous and homogeneous edges. Additionally, it stores the last few selected node domain types to restrict the same node types of accesses. Finally, the node embedding is learned through the SkipGram model;Bidirectional Random Walks are the sampling part of MARU [24]. After this, it is simply called Bidirectional. It involves performing two independent random walks for each node and stitching together a random walk with the node as the center. This is used to overcome the problem that in traditional random walks, low-order nodes are mostly found at the beginning of the walk. It causes the SkipGram model to ignore information at the beginning of the walk, which leads to the loss of node information. Ref. [24] designed a unique heterogeneous SkipGram model and set the window size to the walk length to obtain complete context window information. However, in this paper, only the classical SkipGram algorithm is used for evaluation;SchemaWalk [13] is a schema-aware random walk algorithm for HIN embedding. The edge-sampling mechanism is used to control the selection of edge types by exponential descent. Nodes are then randomly drawn from their corresponding edge types to achieve a more balanced sampling effect. Finally, the sampled random walks are put into the SkipGram model to obtain the corresponding node embeddings and learn the node representations.

Note that DeepWalk was originally designed for homogeneous information networks. Applying it to HIN embedding is to consider the graph as a homogeneous graph. There are also HIN-embedding methods that replace the SkipGram model with a unique heterogeneous SkipGram model, such as HIN2Vec [11], Metapath2Vec++ [9], and MARU [24]. We believe that the most expressive random walks can be combined with different learning methods. In what concerns this paper, we highlight the performance of different random walks as a sampling technique. Therefore, we are using only the classical SkipGram model to learn node embeddings. Methods involving more complex learning components are not used as a baseline because comparisons with them are beyond the scope of this work. 

#### 3.4.3. Implementation

We implement the proposed model HeteEdgeWalk with Python. The Python version is 3.9.16. SkipGram is implemented using Gensim [28], and the Sklearn [29] library is used to evaluate the results. The same hyper-parameters are set for all methods. Specifically, we set walk length l and number of walks λ to 40 and 80, respectively. We also set window size k and dimension size d to 5 and 128, as well as negative samples n to 5. The average of the results of 10 random data segmentation evaluations is used as the final evaluation result. Specifically, for each segmentation, 10% to 80% of the database is utilized as the training set with a 10% step size, and we evaluate it on a 20% testing set.

## 4. Results

In this section, we evaluate the experiments on node classification and node clustering tasks. Subsequently, we analyze the distribution of edge types across random walks and a parameter sensitivity study.

### 4.1. Node Classification Task

The purpose of multi-label node classification is to predict the most probable labels of target nodes according to marked nodes. It focuses on authors in DBLP, movies in Movies, points of interest in Foursquare, and papers in ACM. We train a one-to-all logistic regression function. We divided 80% of the data as the training set and into eight parts, with training percentages [0.1, 0.8] for each split. The remaining 20% of the data served as a testing set. The ten-fold averages of Micro-F1 and Macro-F1 are the evaluation metric. 

Figure 3 illustrates the performance of six random walks on four different HINs for the multi-label node classification. The results show that increasing the training ratio usually leads to better results. First, it is obvious that HeteEdgeWalk achieves optimal or suboptimal results in most cases. The performance in the DBLP is more outstanding. Both HeteEdgeWalk and SchemaWalk achieve high accuracy scores at just 10% of the training percentage, and they demonstrate stable performance on all datasets. Second, since the performance of Metapath2Vec depends on the suitability of meta-path selection for the current dataset, it performs best in Foursquare but worst in Movies. Finally, DeepWalk and Bidirectional perform well on both Movies and ACM. This shows that both datasets are more relevant to correctly classifying nodes without considering node and edge types. However, they do not perform well on datasets with more heterogeneous information. This suggests that both algorithms are more suitable for homogeneous information networks.

To demonstrate the reliability and validity of the method, we perform statistical significance tests on the results of node classification. We analyze the results of Micro-F1 using a paired *t*-test. Our method is an experimental group, with a baseline as a control group.

Table 2 shows the significance level of our method relative to the baseline. The test results show that in most cases, our method is significant at the 0.01 level compared to the baseline. In ACM, our method and Bidirectional did not show differences. Additionally, in Foursquare, Metapath2vec is significantly different and better than our method. This result is also consistent with the node classification results. Overall, all improvements in HeteEdgeWalk are statistically significant at a 95% confidence level relative to the baselines in most cases.

### 4.2. Node Clustering Task

The task is a node clustering with the aim of forming clusters of similar nodes. Specifically, we use the k-means [30] to cluster the node-embedding vectors and normalized mutual information (NMI) as an evaluation metric. We focus on three datasets, DBLP, ACM, and Foursquare, since k-means can only be used for non-overlapping partial clustering, and the labels in Movies are overlapping, i.e., a movie node has multiple labels. This makes the clustering results too low; no experiments are conducted on Movies. The label categories of DBLP and ACM are four and three classes, respectively. In order not to affect the clustering results, we perform a TSNE [31] dimensionality reduction operation on these two datasets before k-means clustering, which reduces the node-embedding vectors to two dimensions. All results are averaged from 10 experimental data. 

Table 3 demonstrates the clustering performance of the three HINs, and it can obviously be viewed that HeteEdgeWalk shows the optimal performance in all three datasets. In all datasets, our method outperforms the other methods by at least 0.6%.

### 4.3. Qualitative Analysis

To allow a comparison of the different sampling results, a qualitative analysis is performed. We display the results by visualization for a more intuitive analysis of the sampling.

Edge Type Sampling Distribution. To analyze the sampling distribution of edge types, we compare the DeepWalk, SchemaWalk, and HeteEdgeWalk methods in DBLP and Movies datasets. Figure 4 displays the distribution of edge types for random walks in doughnut charts. 

In DBLP, the differences between the three random walk methods are evident. The sampling distribution of DeepWalk is skewed because it is biased towards higher-order nodes in the graph, i.e., the more frequently the node type is sampled, the higher the probability. The nodes of type “venue” and their related edge types “paper-venue” and “venue-paper” are sampled much more infrequently than other edge types. SchemaWalk aims to achieve an even distribution of nodes so the overall performance is relatively balanced. However, the homogeneous edges of SchemaWalk “paper-paper” are sampled more frequently than the other types. HeteEdgeWalk has a fairly balanced sampling of heterogeneous edge types. It reduces the sampling frequency of “paper-paper” edge types in comparison to SchemaWalk in exchange for more sampling of heterogeneous edge types. For HIN, the information of heterogeneous edges is more important than that of homogeneous edges. 

In Movies, DeepWalk samples the “actor-actor” with the highest frequency. SchemaWalk is clearly under-sampling “actor-actor” due to the starvation phenomenon mentioned in [13]. The HeteEdgeWalk prefers to sample heterogeneous edge types evenly, with the effect of reduced homogeneous edge sampling being relatively small. In Movies, the distribution of “actor-actor” is unbalanced, with aggregation and edge distribution issues. Aggregation makes it relatively easy to sample the same types of edges here, and the exponential drop too quickly makes some of the same types of edges inaccessible. The edge problem makes the edge types distributed at the edges of the graph, which are relatively difficult to sample and difficult to access. 

It can be seen that our algorithm properly suppresses the sampling of higher-order nodes and homogeneous edges and achieves a more homogeneous sampling of heterogeneous edges. This is also consistent with the theory that it can properly mitigate the bias of higher-order nodes in HIN.

### 4.4. Parameter Sensitivity Study

To investigate the effect of different parameters on the experiment, we conduct a parameter sensitivity study. We investigate the impact of the decay parameter α, the node-embedding dimension d, and the context window k, respectively. 

#### 4.4.1. Impact of the Decay Parameter *α*

The *α* is the decay rate that adjusts the sampling probability for each edge type. We adjust *α* in the range [0.1, 0.9] while holding the other parameters constant in steps of 0.1. Figure 5 illustrates the effect of *α* on the node classification.

The performance peaks at DBLP and Movies when *α* is 0.4 and fluctuates down afterwards. For Movies, it first peaks at *α* = 0.4, and performance improves when *α* > 0.6. In Foursquare, the performance is overall smooth, peaking at *α* = 0.3, then fluctuating down and picking up slightly at *α* = 0.7. In addition to ACM, *α* ∈ [0.2, 0.5] generally gives the finest outcome. This suggests that an appropriate reduction in the value of *α* can help compensate for the edge types that some nodes lack. However, the analysis results of the ACM dataset differ from the other three datasets. We think this has something to do with the ACM dataset itself. Our algorithm mitigates the bias of higher-order nodes in the graph data by applying an appropriate suppression to it. This allows higher-order nodes to be relatively balanced to reach other types of nodes. However, the ACM data itself has only two edge types, A–P and P–S. At *α* > 0.4, first, the increase in sampling of A–P is not enough to compensate for the performance degradation due to the lack of information from the reduction of P–S because the author is relatively more useful to the paper than the subject. After A–P sampling compensates for the loss of information, performance starts to pick up, which may be the reason why ACM is different from the others.

#### 4.4.2. Node-Embedding Dimension d

By fixing other parameters, we increase the dimension d from 4 to 256 to investigate its impact on node-embedding results. Node classification results for the four datasets with different dimensions d are shown in Figure 6a. Firstly, it can be clearly seen that node-embedding cannot capture sufficient graph information when the dimensions are low, resulting in poor performance. In ACM and Movies, the node classification performance increases as the dimension d rises, but the increase tends to be flat. In contrast, in DBLP and Foursquare, performance decreases after reaching a peak. 

#### 4.4.3. Context Window k

We study the effect of the context window k by keeping the other parameters stationary, increasing them from 1 to 10 in steps of 1. The right column in Figure 6b shows the node classification results for four datasets with different k. Firstly, on the whole, a larger k will generally have better node classification performance as it captures the higher-order node nearest to neighboring degrees. In DBLP and ACM, the performance of the node classification results grows sharply as k increases, and then tends to remain stable. In Foursquare and Movies, however, there is a small fluctuation in performance after reaching a peak. There is a fluctuating downward trend in Foursquare.

### 4.5. Visualization 

We use the TSNE [31] dimensionality reduction algorithm to visualize the 2D results of the node embedding in the DBLP and ACM datasets and color them with the ground-truth labels. DeepWalk and SchemaWalk are chosen as the baselines for comparison with HeteEdgeWalk. The upper side of Figure 7 shows the results of DBLP, while the lower side shows the results of ACM. In DBLP, it is clear that the SchemaWalk and HeteEdgeWalk visualizations have four clusters that are more aggregated and consistent with the label set than DeepWalk. The HeteEdgeWalk clusters are the most aggregated and well-defined. However, in ACM, DeepWalk shows some competitiveness, with all three algorithms presenting roughly the same results. Further, the clustering boundaries are relatively fuzzy, which may lead to inaccurate node classification results. The multiple small clustering centers can be seen in DeepWalk, while SchemaWalk has fewer such clustering centers. HeteEdgeWalk performs somewhat better compared to the first two. This result supports the results exhibited by the node classification task.

## 5. Discussion

In this section, we first discuss the overall experimental results. Then, we present some real-world potential applications and future research directions.

The results show a maximum performance improvement of 2% for node classification and at least 0.6% for clustering. This demonstrates, to some extent, the advantages of bidirectional edge sampling. The edge-sampling strategy allows for a more detailed sampling of the graph data. It also proves the advantage of bidirectional random wandering over unidirectional. In the analysis of the sampling distribution of edge types, it is obvious that the state of distribution of sampling for most of the edges accounts for about 14%. This proves that our algorithm can alleviate the bias of high-frequency nodes in HIN and can achieve a more uniform sampling of heterogeneous edges.

Future research could combine other deep-learning algorithms to replace SkipGram for different potential applications. A comprehensive grasp of semantic relations can greatly facilitate the performance of HIN representation learning. Language is also a network of complex relations. Better HIN embeddings are better for the semantic extraction of utterances in natural language processing (NLP) [32]. It can be applied in biology, such as protein interaction networks (PINs) [23] and disease–gene relationship studies [33]. It also has some other applications, such as recommender systems [10], business networks [22], media networks [34], social relationship studies [35], etc.

In addition to static networks, dynamic networks are also an interesting direction for development. Similarity matrices can be constructed for networks of different time periods to study the evolution of dynamic networks [21]. Combining spatiotemporal graphical neural networks (STGNNs) may also be a possible research direction.

## 6. Conclusions

In this paper, the proposed HeteEdgeWalk is a random walk approach for HIN embedding without meta-path guidance. It uses a dynamically adjusted bidirectional edge-sampling walk strategy. To validate the effectiveness of the algorithm, four HINs are evaluated comprehensively in several aspects of node classification, clustering tasks, and edge-type distribution of random walks. The results show a maximum performance improvement of 2% for node classification and at least 0.6% for clustering. A more balanced sampling of heterogeneous edges is achieved in the sampling analysis. The experiments show that HeteEdgeWalk has the advantage. 

The following is a summary of the conclusions: (1) homogeneous graph embedding methods are insufficient to capture heterogeneous graph structures with rich information; (2) sampling based on edge types can capture the rich information in HINs more effectively and carefully than sampling based on node types; (3) HeteEdgeWalk mitigates the effect of high-frequency nodes in random walks and provides a more balanced sampling of heterogeneous edges.

## Figures and Tables

**Figure 1 entropy-25-00998-f001:**
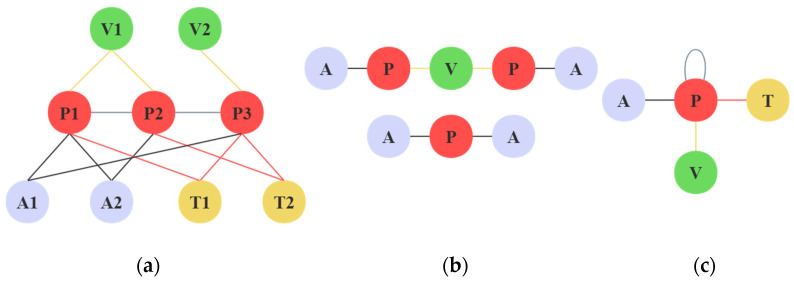
A brief instance of academic Heterogeneous Information Network. (**a**) Heterogeneous Information Network. (**b**) Meta-paths. (**c**) Network schema.

**Figure 2 entropy-25-00998-f002:**
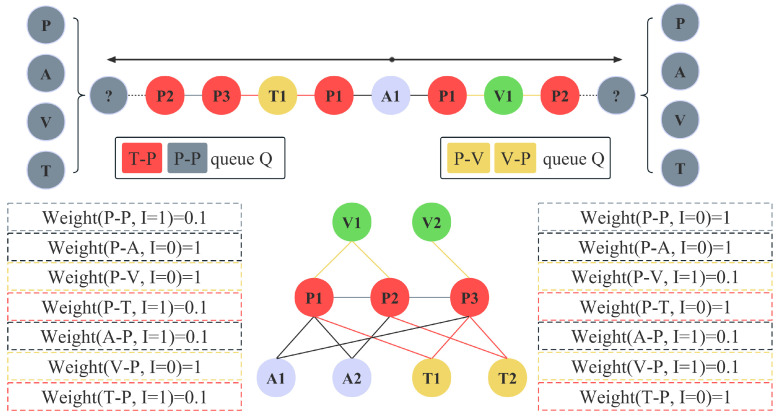
The instance of Bi-EdgeSamplingWalk on academic graph, where *α* is set to 0.1. The walker selects the next edge type at node *P*2.

**Figure 3 entropy-25-00998-f003:**
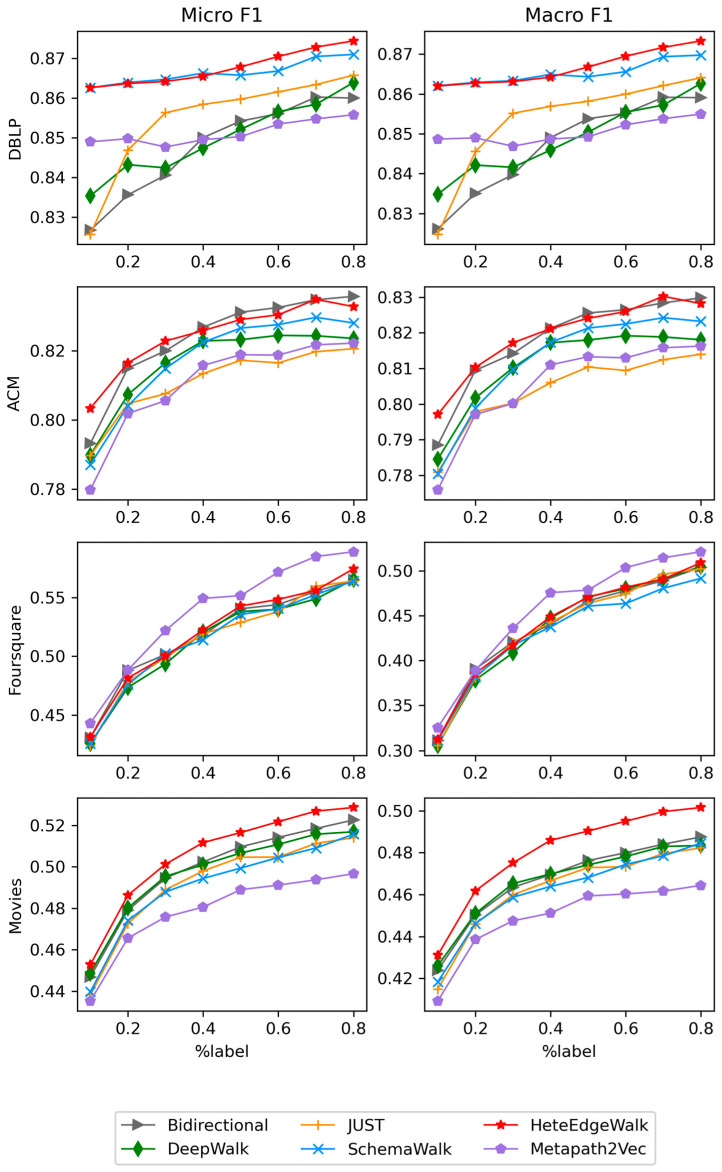
Performance of node classification task.

**Figure 4 entropy-25-00998-f004:**
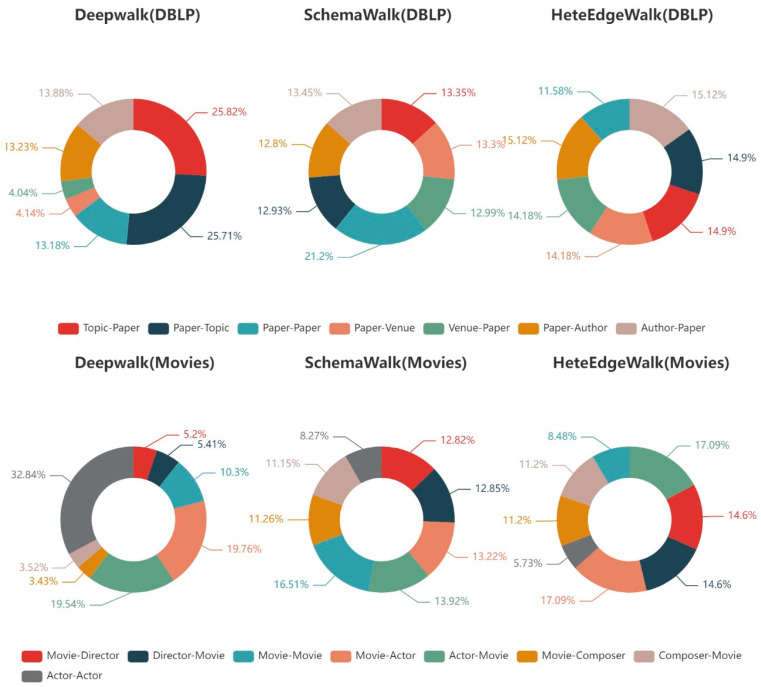
Doughnut charts of the distribution of edge types for random walks.

**Figure 5 entropy-25-00998-f005:**
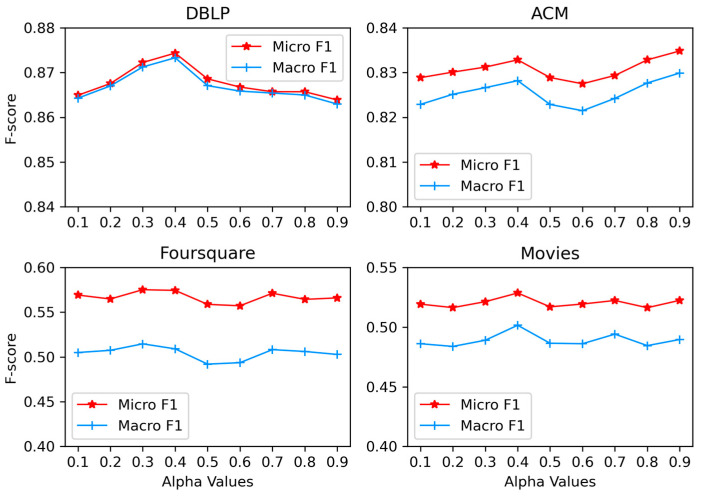
Impact of the decay parameter *α*.

**Figure 6 entropy-25-00998-f006:**
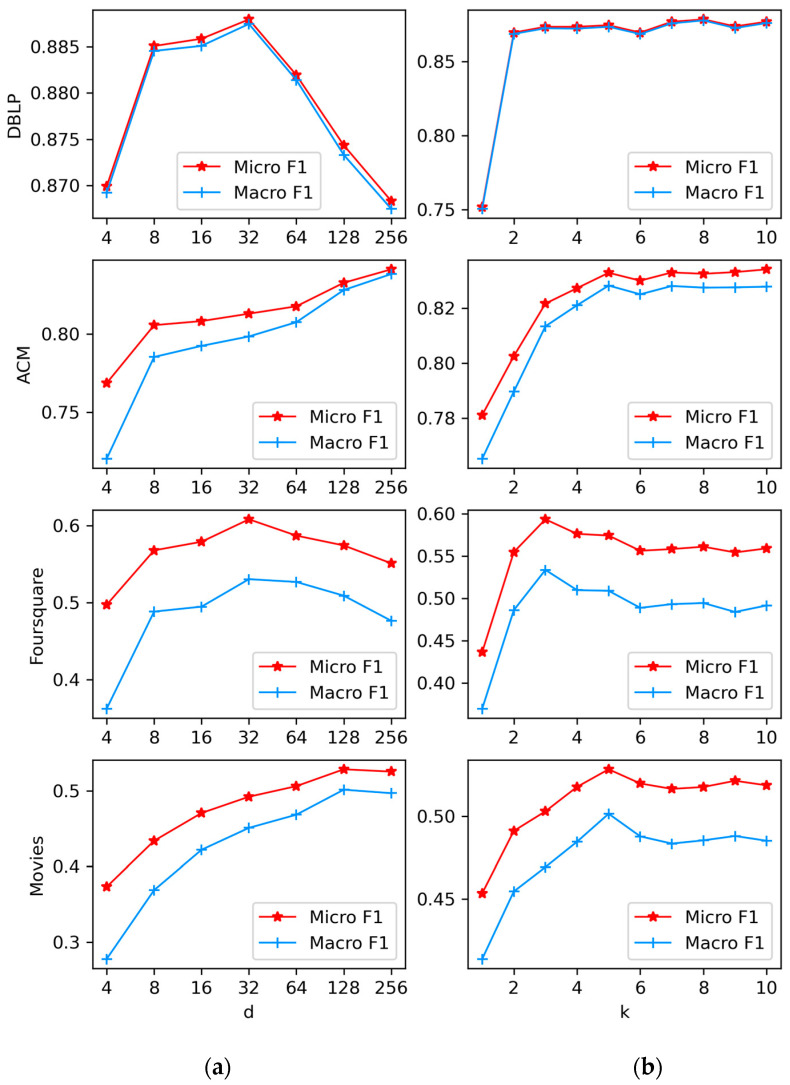
Influence of other parameters. (**a**) The impact of parameter d. (**b**) The impact of parameter k.

**Figure 7 entropy-25-00998-f007:**
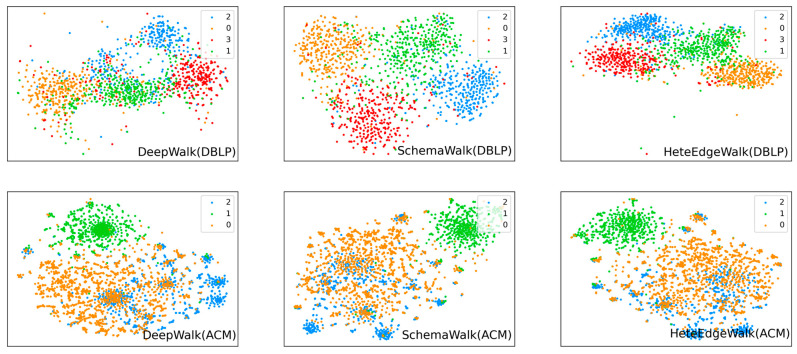
The 2D visualization of the node embeddings.

**Table 1 entropy-25-00998-t001:** The statistics of the experimental HIN.

Dataset	Node	Edge
DBLP	author: 5915	P-T: 26,532
	paper: 5237	P-A: 13,589
	topic: 4479	P-P: 6984
	venue: 18	P-V: 4258
ACM	author: 7167	A-P: 13,407 P-S: 4019
	paper: 4039	
	subject: 60	
Foursquare	check-in: 25,904	U-C: 25,904
	user: 2449	C-P: 25,904
	point of interest: 1250	C-T: 25,904
	timestamp: 168	U-U: 5696
Movies	actor: 10,350 movie: 6517 director: 2582 composer: 1335	A-A: 43,395 M-A: 23,223 M-M: 6194 M-D: 5630 M-C: 4001

**Table 2 entropy-25-00998-t002:** The statistical significance test results for paired *t*-tests.

Method	DBLP		ACM		Foursquare		Movies	
	*t*	*p*	*t*	*p*	*t*	*p*	*t*	*p*
DeepWalk	−9.474	0.000 **	−6.787	0.000 **	−7.454	0.000 **	−8.399	0.000 **
Metapath2vec	−25.272	0.000 **	−8.457	0.000 **	5.952	0.001 **	−13.641	0.000 **
JUST	−3.614	0.009 **	−26.185	0.000 **	−2.798	0.027 *	−23.956	0.000 **
Bidirectional	−6.554	0.000 **	−0.535	0.609	−3.792	0.007 **	−17.626	0.000 **
SchemaWalk	−8.670	0.000 **	−3.904	0.006 **	−4.198	0.004 **	−17.608	0.000 **

* *p* < 0.05, ** *p* < 0.01.

**Table 3 entropy-25-00998-t003:** The NMI for node clustering.

Method	DBLP	ACM	Foursquare
DeepWalk	0.121	0.364	0.281
Metapath2Vec	0.551	0.371	0.288
JUST	0.136	0.341	0.287
Bidirectional	0.248	0.375	0.275
SchemaWalk	0.593	0.322	0.271
**HeteEdgeWalk**	**0.599**	**0.381**	**0.294**

## Data Availability

Publicly available datasets were analyzed in this study. Foursquare is a graph based on users’ check-in history in New York city that is one of 2535 cities in ESRI http://www.esri.com. We use a subset of DBLP in November 2009, i.e., DBLP-4-Area https://dblp.org/xml/release/. Movies is the subset from CORE Facts of YAGO https://www.mpi-inf.mpg.de/departments/databases-and-information-systems/research/yago-naga/yago/. ACM is the subset http://dl.acm.org/.

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
