# Peer review of "HeteEdgeWalk: A Heterogeneous Edge Memory Random Walk for Heterogeneous Information Network Embedding"

_entropy, 2023, doi:10.3390/e25070998_

Round 1

Reviewer 1 Report

The motivation and objective of the presented work look good.

The abstract needs to be modified adding key results obtained through simulations/experiments. The Related Work section looks good, but the reviewer believes that the author can improve this section adding more related work available in the literature and show the true contributions of the presented work in comparison to and in line with the state-of-the-art research work. The Experiments in Section 4 should be further described. The authors mentioned the data sets, but the experimental procedures were not explained adequately. Experimental results should be presented more clearly in a separate section “Experimental Results”.  

The authors presented a lot of plots. However, the authors should further explain the plots and discuss how the results prove the effectiveness of the proposed algorithms. A discussion section may be added. The conclusions section should be improved by adding key results that the authors obtained in their experiments and explaining the importance of the results. The future direction of the research should be mentioned.

Minor editing of English language required.

Reviewer 2 Report

The authors of the submitted paper, entropy-2456549, presents a novel method of graph embedding for heterogeneous information networks (HINs). The authors propose a flexible approach, named HeteEdgeWalk, that leverages random walks to overcome the limitations of meta-path based strategies that currently dominate the field. The paper is technically sound and methodologically rigorous, with a clear structure. The introduction provides a comprehensive overview of the problem space and sets the stage for the authors' novel approach. The presentation of the HeteEdgeWalk method is clear and systematic. The authors provide well-explained mathematical models and algorithms, shedding light on the specifics of their approach. The experiments are thorough, and the results are presented clearly. The authors conduct both quantitative and qualitative analyses to demonstrate the effectiveness of HeteEdgeWalk. The examination of multiple variables (decay parameter α, node embedding dimension d, context window k) under various conditions provides substantial evidence supporting their claims. It is commendable that they included comprehensive sensitivity analyses, adding credibility to their results. One of the significant strengths of the paper is the comparative analysis against other methods, such as DeepWalk and SchemaWalk. The comparative analysis against both homogeneous and heterogeneous network methods provides a balanced view. The visualization of node embeddings offers an intuitive understanding of the data, which is a significant advantage for this kind of work. The authors have shown the results of their method and compared it to existing methods, which enhances the reader's comprehension. However, there are some aspects that could be improved upon. For instance, the authors may want to delve deeper into why certain trends were observed in the experiments. The explanation regarding the decay parameter α's impact, for example, could be expanded upon. The unusual performance on the ACM dataset, which is more homogeneously explored, warrants further exploration. Furthermore, while the authors mention that the HeteEdgeWalk method theoretically mitigates bias in random walks and HINs, they could benefit from providing more direct empirical evidence of this bias mitigation in their results. Lastly, the paper could be enhanced by a discussion on the potential practical applications of HeteEdgeWalk and possible future enhancements. It would be interesting to understand how the new method could be applied in real-world scenarios and its impact on different domains. Overall, this paper makes a significant contribution to the field by proposing a novel and promising method for HIN embedding. The research is of high quality and potentially paves the way for advancements in this field. The proposed enhancements should in no way detract from the quality of the research but are suggested to add further depth and insight into their work.

General comments:

§  The paper presents a novel and potentially significant contribution to the field of heterogeneous graph embedding.

§  The introduction, problem description, and presentation of the HeteEdgeWalk method are well-structured and clearly articulated.

§  Comprehensive experiments are conducted, and results are presented systematically, providing strong evidence to support the proposed method.

§  The comparative analysis against other methods, including both homogeneous and heterogeneous network methods, offers a balanced perspective.

§  The visualization of node embeddings adds substantial interpretability to the research.

§  Some trends observed in the experiments could benefit from deeper analysis or explanations (e.g., impact of the decay parameter α).

§  The paper could be enhanced by more direct empirical evidence of bias mitigation in random walks and HINs.

§  The paper could further discuss potential real-world applications of HeteEdgeWalk and future enhancements.

Questions for authors:

§  Could the authors provide additional insights into why the decay parameter α performs differently for the ACM dataset compared to others?

§  Are there potential ways to optimize the computational cost as the complexity and size of the graph increase?

§  How does HeteEdgeWalk perform when dealing with highly unbalanced graphs where some types of nodes or edges dominate?

§  Can the authors discuss more real-world applications where HeteEdgeWalk could be applied and provide examples of how it might perform better than existing methods in these scenarios?

§  Can HeteEdgeWalk handle dynamic graphs where nodes and edges can be added or removed over time? If so, how would it adapt to the changes?

§  Could the authors clarify the potential for overfitting with HeteEdgeWalk, particularly when dealing with sparse or noisy datasets?

§  Could the authors further elaborate on how HeteEdgeWalk mitigates the bias of tails in random walks and the different types of bias in HINs with empirical evidence?

§  How does the choice of the dimension d influence the quality of the embeddings in different kinds of datasets? Can the authors provide a rule-of-thumb or guidelines for selecting this parameter in various contexts?

§  The authors mentioned the use of the SkipGram model for learning the node embeddings. Could they elaborate on why they chose this model over others and if there could be potential benefits in experimenting with different models?

It seems that the paper is relatively well written in terms of English language usage. The sentences appear coherent and correctly structured, with appropriate use of technical language and terminology. However, there are a few minor issues that could be improved for better readability. Some sentences are quite long and complex, which might challenge the reader's understanding. Breaking these down into simpler sentences could improve clarity. Some of the technical terms and abbreviations (e.g., HIN, ACM, DBLP) are used without a full explanation or context. Although these might be well-known within the field, providing a brief description could make the paper more accessible to a broader audience. The authors could improve the transitions between some points and sections for a smoother reading experience. While the overall quality of English appears good, minor revisions could improve the readability and accessibility of the paper.

Reviewer 3 Report

In this paper, authors proposed HeteEdgeWalk, a Heterogeneous Information Network embedding methodology, based on network schema.

Introduction and related works are exhaustive. I suggest also mentioning this work: PMID:37190452.

I suggest using “Materials and Methods” as title for Section 3, and “Results and Discussion” for Section 4, in accordance with the well-known “Introduction Method Results Discussion” (IMRaD) organization.

The method (Section 3) is well explained, formally. However, I suggest better commenting on the various passages, or reporting a report at the end of the section related to Algorithm2, in order to guide the reader in understanding the workflow.

Results and Discussion support your work, as well. However, statistical significance tests are missing. These could be useful to prove the reliability and effectiveness of your methods.

In my opinion, the manuscript addresses a very interesting topic, and its application is relevant for bioinformatics. 

The way of writing is confusing and should be simplified, by reducing the fragmentation of sections 3 and 4, as much as possible. In addition, I suggest checking for typos.

Round 2

Reviewer 3 Report

Authors improved the manuscript by addressing my suggestions. The overall merit may be considered as relevant for the journal, therefore, I suggest accepting the manuscript in present form.

Checking for typos.